# Electrical Resistance Tomography for Control Applications: Quantitative Study of the Gas-Liquid Distribution inside A Cyclone

**DOI:** 10.3390/s20216069

**Published:** 2020-10-25

**Authors:** Muhammad Awais Sattar, Matheus Martinez Garcia, Robert Banasiak, Luis M. Portela, Laurent Babout

**Affiliations:** 1Institute of Applied Computer Science, The Lodz University of Technology, Stefanowskiego 18/22, 90-924 Łódź, Poland; robert.banasiak@p.lodz.pl (R.B.); laurent.babout@p.lodz.pl (L.B.); 2Department of Chemical Engineering, Delft University of Technology, Van der Maasweg 9, 2629 HZ Delft, The Netherlands; m.martinezgarcia@tudelft.nl (M.M.G.); l.portela@tudelft.nl (L.M.P.)

**Keywords:** electrical resistance tomography (ERT), swirling two-phase flow, digital image processing, high-speed camera

## Abstract

Phase separation based centrifugal forces is effective, and thus widely explored by the process industry. In an inline swirl separator, a core of the light phase is formed in the center of the device and captured further downstream. Given the inlet conditions, this gas core created varies in shape and size. To predict the separation behavior and control the process in an optimal way, the gas core diameter should be measured with the minimum possible intrusiveness. Process tomography techniques such as electrical resistance tomography (ERT) allows us to measure the gas core diameter in a fast and non-intrusive way. Due to the soft-field nature and ill-posed problem in solving the inverse problem, especially in the area of low spatial resolution, the reconstructed images often overestimate the diameter of the object under consideration leading to unreliable measurements. To use ERT measurements as an input for the controller, the estimated diameters should be corrected based on secondary measurements, e.g., optical techniques such as high-speed cameras. In this context, image processing and image analysis techniques were adapted to compare the diameter calculated by an ERT system and a fast camera. In this paper, a correction method is introduced to correct the diameter obtained by ERT based on static measurements. The proposed method reduced the ERT error of dynamic measurements of the gas core size from over 300% to below 20%, making it a reliable sensing technique for controlled separation processes.

## 1. Introduction

The continuous rise of computational power in recent years resulted in faster acquisition and reconstruction of tomographic measurements, opening the door to the monitoring of industrial processes in real-time by the technique. In such a context, the concept of including the rich internal data provided by the sensor for real-time control of industrial processes has arisen [1].

Soft-field electrical tomographic techniques, such as Electrical Capacitance Tomography (ECT) and Electrical Resistance Tomography (ERT), are widely used as a non-intrusive/non-destructive measurement technique to monitor processes such as pneumatic conveying/granular flow in silos [2,3,4,5,6], mixing processes [7,8], batch crystallization [9,10], geophysical surveys [11] and two-phase flows [12,13] to name a few. Research on ERT as a non-intrusive method for two-phase flow measurements has been carried out since the last few decades, primarily due to the higher measurement speed, low implementation cost, and straightforward implementation [14].

Electrical Resistance Tomography can provide valuable measurements for the control of an axial cyclone (swirl tube/inline swirl separator) by evaluating in real-time the distribution of the mixture inside the device. Previous studies [15,16] have been done in swirling two-phase flows. However, the literature lacks in providing a fast and reliable way to estimate void fractions for real-time control applications.

To illustrate the use of tomography in the context of an inline swirl separator, a sketch of the typical setup is presented in Figure 1.

The working principle behind a swirl separator is the centrifugal force. When a mixture of two phases of different densities is pushed through the swirl element, angular velocity is created, and centrifugal forces take place, forcing the phase with higher density toward the pipe wall and creating a core structure with the less dense medium in the center [17]. The efficiency of the separation is determined by the location of the fluids (phases) at the entrance of the two outlet ducts [18]. 

Convection plays a vital role in the flow, and changes in the concentration of fluids crossing the beginning of the separator, close to the swirl element, will reach the outlets after a while. The future separation can be predicted by combining the measurement of the concentration of each fluid at the ERT sensor with a model of the process. When the connection is established, controllers can be designed to regulate the separation by acting on the control valves present in the outlets.

Another possible application of the tomographic measurement in control would be the monitoring of the separation taking place close to or inside the outlet pipes, providing feedback on the process. In both scenarios, reliable and fast measurement of the multiphase flow is fundamental for the proper operation of a controller connected to the device.

ERT based sensors are used to obtain conductivity distribution of the media under observation, and this can be achieved by placing sensing electrodes in contact with the targeted medium [19]. Due to considerable conductivity differences between air and water phases, the gas core can be efficiently detected using ERT. A typical ERT system comprises of ERT sensor, a Data acquisition system, and image reconstruction, and visualization software [20], as shown in Figure 2.

For data acquisition, ERT uses different measurement methods such as Adjacent Drive Method (ADM) [21], Opposite method [22], and Trigonometric method [23], whereas ADM is the most commonly used measurement method.

The ERT instruments may be classified into current-voltage (CV-ERT) (or current mode) and voltage-current (VC-ERT) (or voltage mode) systems. In the former, currents are injected through small area electrodes, and voltages are measured. In the latter, voltages are applied (injected) to relatively large-sized electrodes (similar to the one used in ECT systems), and currents are measured. At the beginning of ERT development, a pioneer VC-ERT system was developed by Henderson and Webster (1978) [24]. They designed an impedance camera to test the feasibility of impedance imaging of the thorax.

Due to unknown contact impedances, the CV-ERT based framework can generate good quality results when compares to VC-ERT based systems [25]. Furthermore, the VC-ERT framework provides an increase in sensitivity to electrode placement and size errors; thus, a better quality of images can be generated [26]. Considering the advantages, VC-ERT systems now are widely being used in worldwide medical research groups [27,28,29,30,31]. However, the mathematical formulation of the forward solution in VC-ERT follows the classic CV-ERT principle. The partial differential equation derived from the Maxwell equation is used to define the relationship between the conductivity distribution and the boundary voltages [32]. The only difference, as detailed in [33], relies upon the expression of the conductance tensor ***G***, which relates the injected voltages and measured currents, and is the opposite value of the estimated conductance for the CV case, i.e.,:(1)$$G_{VC}=-G_{CV}=D-C^{T}B^{-1}C$$
where the tensors ***B***, ***C***, and ***D*** are discretized solutions of the FEM complete forward model, which depends on the geometry of the problem domain (e.g., electrodes size, dimensions of the area of investigation, number of mesh elements) and conductivity distribution.

When using a pair of voltage excitation patterns, the data collection method of the VC-ERT system should be different from that of the CV-ERT system. In the VC-ERT system, a single voltage source can excite multiple electrodes at the same time. The VC-ERT system introduces a new single-source pattern different from pair-excitation patterns such as adjacent and opposite methods are common in the CV-ERT system. When a voltage signal is injected between $e_{1}$ electrode (source electrode) and the rest of the electrodes (sink electrodes), current data are measured successively ($I_{1}$, $I_{2}$, …, $I_{15}$ For N = 16) from all the electrodes, excluding the voltage-injecting electrode. Next, the second voltage pattern is applied through $e_{2}$ and ($e_{3}$, $e_{4}$, …, $e_{16}$, $e_{1}$), and then-current data are measured successively. Therefore, for a 16-electrode VC-ERT system, this method of data collection gives M = N(N − 1)/2 = 120 electrode pairs measurement.

Image reconstruction is done by processing the acquired data to determine electrical conductivity distribution. Depending upon the change in electrical conductivity, the inverse problem can be explained considering a linearized system of equation [34]:(2)δV=J·δσ
where *J* is known as a Jacobian matrix, i.e., the matrix of sensitivity distribution and can be computed by analyzing the change in the current *λ* for each conductivity $\sigma_{n}$ [33]:(3)$$J=\frac{\partial \lambda }{\partial \sigma_{n}}$$

The goal of this paper is to evaluate the Electrical Resistance Tomography measurement in terms of accuracy and reconstruction speed when evaluating gas-liquid swirl flows for control purposes. The reconstruction routines adopted were chosen to target at a fast (non-iterative) reconstruction scheme, and the ERT values observed are compared to flow imaging using a high-speed camera to evaluate its response. The main novelty of the work resides in the experimental protocol and the camera-based correction approach to calibrate the ERT-based estimation of gas core diameter during dynamic tests of inline fluid separation.

This work is part of an EU Horizon 2020 project name TOMOCON (Smart tomographic sensors for advanced industrial process control which aims to integrate imaging sensors into industrial processes. This manuscript is arranged in the following manner. Section 2 focuses on the flow test facility, sensors, and image processing techniques used in this work. Section 3 describes the dynamic and static experimentation results in detail, and Section 4 presents the article’s conclusion.

## 2. Materials and Methods

### 2.1. Flow Test Facility and Phantoms

The experiments were performed using the two-phase liquid/gas flow facility of the Tom Dyakowski Process Tomography Laboratory at the Lodz University of Technology (TUL). The liquid/gas flow facility consists of horizontal and vertical flow loops of three different pipe dimensions (outer diameters: 90 mm, 50 mm, and 30 mm), as shown in Figure 3a. The flow loop consists of a buffer tank, followed by a centrifugal pump and a flow meter. Tap water of electrical conductivity 60 mS/m [35] was used in the installation. A compressor of 5 bar capacity was used to produce gas, which was injected in the flow installation using the Brooks Mass Flow Controller (MFC). The valve operations and Brooks control were carried out using National Instruments (NI) system. The experiments were conducted using the vertical pipe of the 90 mm outer diameter, as shown in Figure 3b.

The installation was modified in a modular manner such that different sensors could be tested without causing harm to the rest of the structure. The swirl element was placed on the 90 mm outer diameter (84 mm internal diameter) pipe and 1 m above the water injection point. A high-speed camera from Basler (model: acA2000-165 um USB 3.0 with CMOSIS CMV2000 CMOS sensor) was used to record the flow at 50 fps. An artificial light source was used to improve the amount of light reaching the camera sensor at low shutter speeds. It consists of 2 fluorescent lamps (model: Spectralux Plus Radium NL 18 W/840, 1300 Lumen) installed in parallel installed in the back of the wood box of Figure 3b, covered by a sheet of white paper to diffuse the light. The flow is recorded by the camera at the level of the ERT sensors and compared with the tomography to check its accuracy. Twenty-five experimental points were chosen to study the gas core based on the ERT sensor and the camera. They were chosen based on steps of 5 m^3^/h in the flow rates of each phase, starting at 10 m^3^/h. The air values were selected at standard conditions, as provided by the MFC, and thus must be corrected based on pressure values measured right after the mass flow controller. The approach results in the experimental points presented in Table A1
Appendix A, where the superficial velocity of the fluid “j” corresponds to the bulk velocity of the phase if it was the only fluid in the pipeline:(4)$$V_{s\;j}=\frac{q_{j}}{A}$$

In the expression, $V_{sj}$ corresponds to the superficial velocity of the phase “*j*” $q_{j}$ to the flow rate of the phase “*j*” in m^3^/s, and A to the pipe area ($D_{in}=84\;\mathrm{mm},\;A=5.54\times 10^{-3}$ m^2^).

To better illustrate the velocities chosen, the superficial velocity pairs are plotted in the Taitel et al. [36] mechanistic map for vertical gas-liquid non-swirling flows shown in Figure 4. The churn-slug transition is not plotted in the map, as the 1 m (≈$12\;D_{in}$) between the Swirl Element and the gas injection is not sufficient for the development of a slug flow for the velocity range studied according to the article’s theory. Figure 4 indicates that the experimental points are situated around the bubble-churn transition, although no verification of the patterns was performed during the experiments.

A set of flow images from the fast camera were recorded at 50 Hz, while the flow measurements obtained from a single layer 16-electrode ERT sensor placed 500 mm downstream of the swirl element were acquired at 12 Hz. During each experiment, 1 min of ERT raw data was recorded using the TomoKisStudio software [37]. Due to the memory occupied by the fast camera images, only the last 10 s were captured, always starting when the ERT reaches about 600 measurements.

Prior to dynamic testing, static measurements were performed to analyze the accuracy of the camera and ERT reconstruction algorithms described below. Non conducting phantoms of known diameters ranging from 10 mm to 40 mm were designed to mimic the gas core. Blender v2.79 modeling software suite was used to create the phantoms. Ultimaker-3D printer was used to print these phantoms using Acrylonitrile Butadiene Styrene (ABS) material.

### 2.2. ERT Sensor snd Measurement Protocol

The physical sensor is shown in Figure 5a. It consists of 16 stainless steel electrodes of circular shape. Each electrode has a diameter of 12 mm head and a 5 mm thread. The electrodes were placed equidistantly on the inner surface of the plexiglass pipe of 90 mm outer and 84 mm diameters by drilling holes. The holes were sealed using rubber sealing of 2 mm thickness. A distance of 5.2 mm was kept between each electrode to eliminate the harm of cross talk. The system used for this research was the Flow Watch device from Rocsole Ltd. The protocol of measurement of the Flow watch system is voltage injection and current measurement (VC) (Figure 5b), as described in the Introduction section.

The frame rate for acquiring the data was 12 Hz. The sensor is connected to the data acquisition electronics using signal conditioning units, which can amplify or de-amplify the voltage signal depending upon the target media’s conductivity. Coaxial RF connectors (MCX) were used for the connection between measurement electronics and the sensor.

As far as the experimental protocol is concerned for both static and dynamic measurements, the sensor was filled up to the electrode level by Tap water (500 mL). Before taking the ERT measurement, reference measurements were performed with just water inside the sensor. The purpose of the Swirl element is to align the gas core right in the center of the pipe, and to mimic its behavior, the designed phantoms were placed in the center of the sensor, as shown in Figure 6.

### 2.3. Image Reconstruction Algorithms

The main objective of ERT is to solve a nonlinear problem, but for simplification, a linear approximation was assumed. The following equation can describe the conductivity distribution for measured currents:(5)$$\sigma =J^{-1}\lambda$$

As *J* is an ill-posed matrix, so there exists no direct solution; thus, we need to solve it using inverse solutions methods. For this research, the Linear Back Projection (LBP) and one-step Gauss-Newton (GN) methods were used, even though other ways have been proposed to better estimate the electrical property distribution of binary mixture, including iterative approaches [34], machine learning [38] or fast Jacobian matrix inversion-based method [39].

LBP is a fast, classical, and simple image reconstruction algorithm with many applications in multiphase flow measurements [40,41]. LBP can modify Equation (5) as in [42]:(6)$$\sigma \approx S^{T}\lambda_{m}$$
where $S^{T}$ represents the transpose of the Jacobian matrix and $\lambda_{m}$ represents the measured current vector. However, the GN solving method solves the inverse problem in terms of generalized Tikhonov regularization [43]. GN solving approach provides a more stable and reliable solution and can be described as:(7)$$\sigma ={\left(S^{T}\cdot S+\alpha U\right)}^{-1}[S^{T}\left(\lambda_{m}-\lambda_{pr}\right)]$$
where α is the regularization parameter which can be computed experimentally, *U* is the unity matrix and $\lambda_{pr}$ is the predicted current measurements computed from the forward problem.

In this study, the difference imaging (DI) technique was used using both LBP and GN methods. In the DI method, the sensitivity matrix is pre-calculated for the sensor geometry using the forward model data, and it is used to get a reference image from the measurements [42]. The difference conductivity vector can be described as:(8)$$\sigma_{diff}=\sigma_{inh}-\sigma_{h}$$
where $\sigma_{h}$ is the homogenous absolute conductivity, which represents the liquid phase in this case and $\sigma_{inh}$ is the inhomogeneous absolute conductivity representing the gas phase.

Image reconstruction was done using EIDORS Version 3.10, which is open-source software for Diffusion based Optical Tomography and Electrical Impedance Tomography (EIT) for industrial and medical applications [44]. Most of EIDORS functionalities and applications can be found in [45]. For electrode placements, sensor geometry and FEM meshing NETGEN [46] were used together with EIDORS. The image reconstruction procedure followed by EIDORS is shown in Figure 7. For the image reconstruction, CPU with Intel^®^ Core™i7 1.80 GHz, 16 GB installed memory (RAM), and 64-bit Windows 10 operating system was used.

### 2.4. Image Processing

Void fractions were calculated from both ERT and fast camera images using image processing algorithms. Matlab^®^ R2019b Image Processing Toolbox™ was used for the postprocessing of images obtained by both modalities.

#### 2.4.1. ERT Image Handling

The main objective of this image handling step is to estimate the void fraction, which is a primary estimation of the liquid-gas distribution in the cyclone. In that matter, two different classical approaches were studied in order to distinguish the better of the two, that is:Averaging the grayscale images.Image segmentation of the images

In medium distributed images of gas-liquid swirling multiphase flow, there exists a recognizable interface between two phases, which can be noticed by observing the grey level changes at the phase border [47]. The image reconstruction algorithm generates an image of 344 × 343 px. The images were transformed into grayscale using the *rgb2gray* function. First, the total number of pixels representing the inner diameter of the pipe was computed. Then with the help of the Matlab script, the total number of light and dark pixels [48] were computed, and using Equation (9), the void fraction was computed. An advantage of the average grayscale image schemes is that they don’t depend on image processing techniques as binarization, which depends on finding an optimal threshold value.

The second tested approach consists of a semi-automatic image segmentation technique called the Graph cut method (GCM) [49,50]. GCM is used to label the image into the foreground and background. Thus, the gas phase can be regarded as a foreground and the liquid phase as a background. The segmented image was further processed with operations like border clearing *imclearborder* and holes filling *imfill* to improve the segmentation quality. The whole procedure of image processing is described in Figure 8.

The cross-sectional void fraction from both the methods can be calculated using the following equation [51]:(9)$$\alpha =\left(1-\overset{M}{\underset{j=1}{\sum }}f_{j}\frac{A_{j}}{A}\right)$$
where $A_{j}$ is the area of the *j*th pixel, A represents the total area of the sensing area, *M* is the total number of pixels in the reconstructed image and $f_{j}$ is the normalized calculated level (gray or binary) of the reconstructed image.

#### 2.4.2. Fast Camera Image Processing

The images were recorded during the last 10 s of the ERT measurements, resulting in 500 frames captured per experimental point. As the core presents random oscillations of size in both measures, and the camera is triggered manually, it is not possible to accurately synchronize the ERT and camera signals when evaluating the measurements.

To improve the lighting of the images, and thus the contrast created at the interface between air and water, two fluorescent lamps are used in parallel to illuminate the back of the pipeline while recording it. The fluorescent lamps cause a drift of the background color of the stack of images over time, which effects are minimized during the pre-processing of the camera pictures. The complete image processing procedure is explained in the following subsections.

##### Pre-Processing Routine

Based on the apparent location of the inner wall of the pipe, the obtained flow pictures are carefully cropped in a rectangular box of size 200 × 400 px, starting one outer diameter after the ERT layer of electrodes. The box was chosen such that the location is sufficiently small, and the average values calculated still represent the dynamical behavior observed by the ERT. However, at the same time, the number of pixels is enough to ensure some stability to the gas core tracking errors by the image processing algorithm. The cropping of the image is illustrated in Figure 9.

For each experimental point, the set of cropped images is broken into stacks of 50 images (corresponding to 1 s of recording) to select regions where the background color is approximately the same. The images are then converted from RGB to 8-bit grayscale. In the software, the experimental data is broken into stacks of 50 images (due to the drifting in the background color by the fluorescent lamp), the maximum intensity projection of the stack is subtracted from the pictures, and the resulting images have their contrast-enhanced. Figure 10 presents the original box cropped from the flow pictures and the result of the pre-processing routine.

##### Gas Core Calculations

The calculation of the core properties is made in MATLAB. Initially, each image is binarized via the *imbinarize* function, with the threshold calculated by the Otsu method [52]. Then, the colors of the image are inverted via the *imcomplement* function, which is required for the removal of small elements via the *bwareopen* function.

The *bwareopen* function is set to remove small white regions, now representing the interfaces, of size less than 100 px. The region sizes are calculated considering 8-element connectivity, i.e., the pixels share the same region if they have the same value and intersect each other (on any vertex or edge). The procedure is required to eliminate bubbles that are far from the core and small residuals of the binarization step.

A gas core is expected in the images from the flow physics. Then, a for-loop is written to estimate the gas core diameter for each column of pixels inside the image by searching for the maximum radial position where the value of 1, representing the interface, is followed by 0, representing the water annulus. The image processing steps using MATLAB are illustrated in Figure 11, with the reconstructed core interface represented in red in Figure 11c.

Once the interface position over the length of an image is obtained, a routine to correct each point (pixel on the interface) is performed based on the refraction suffered by the light rays reaching the camera.

##### Refraction Correction

The filmed images present a distortion in the relation of the real flow inside the pipeline due to refraction in the transitions between materials, being significantly affected by the curved wall of the pipeline. To better approximate the gas core interface position observed in the images to the real location taking place inside the pipe, the path of the light rays being reflected by the gas-liquid interface is studied based on [53], according to the schematics of Figure 12.

Following Snell’s law, the change in the light ray angle at the interfaces between the different materials is given by:(10)$$n_{air}\sin\phi_{1}=n_{acrylic}\sin\phi_{2}$$
(11)$$n_{acrylic}\sin\phi_{3}=n_{water}\sin\phi_{4}$$

Corresponding to the refractions at points B and C of Figure 12, respectively. In the equation, $n_{i}$ corresponds to the refractive index of the materials and $\phi_{i}$ to the angles measured in relation to the normal direction of the transitions. In this article, it is considered $n_{air}=1$, $n_{water}=1.333$ (λ=589 nm, T=20 °C [54]), and $n_{acrylic}=1.493$ (λ=589 nm,  T=20.1 °C) [55].

Before departing to the geometrical relations that close the current set of equations, and allow the correction on the core position to be performed, there is a critical hypothesis that needs to be made: the interface is assumed to be located precisely at the plane crossing the center of the pipeline and parallel to the view of the camera, i.e., on top of the *x*-axis of Figure 12.

The assumption is required because there is no distinction in the reflected light ray path for any point contained in the straight line segment $\bar{{\mathrm{CD}}^{\prime }}$ [53]; they would all reach the same pixel of the picture, with slight differences in focus. As the gas core formed by the swirl motion is expected to take place in the central region of the pipe, and an average of the values obtained is performed, the simplification seems reasonable.

Finally, from the geometry of the light ray represented in Figure 12, it is possible to write:(12)$$\Omega -\phi_{1}+\phi_{2}-\phi_{3}+\phi_{4}+\gamma =\frac{\pi }{2}$$
(13)$$\frac{\sin\gamma }{R_{in}}=\frac{\sin\phi_{4}}{x_{real}}$$
(14)$$\frac{\sin\phi_{2}}{R_{in}}=\frac{\sin\phi_{3}}{R_{out}}$$
(15)$$\frac{\sin\phi_{1}}{L}=\frac{\sin\Omega }{R_{out}}$$

Corresponding to the sum of internal angles of the pentagon OABCD = 3π (Equation (11)) and the law of sine of the triangles OCD, OBC, and OAB (Equations (12)–(14) respectively). In the equations, Ω corresponds to the angle between the apparent interface location ($x_{apparent}$) and the camera, γ is an additional angle that simplifies the relations, L is the distance between the camera and the pipe centerline, $R_{in}$ and $R_{out}$ are the pipe inner and outer radii, respectively, and $x_{real}$ is the “real” location of the interface (considering the hypothesis of the interface points being on top of the pipe center plane parallel to the camera view).

The image captured by the camera is assumed to be a perfect miniature of the reality, such that the distances observed, in meters, are proportional to the distances, in pixels, of the image ($x_{apparent}=k_{1}x_{px}$). A convenient point to set the scaling factor between the picture and the reality is the outer wall of the pipeline, where $x_{apparent}=x_{real}$ (due to the lack of refraction effects), and both positions are known (45 mm in reality and 205±1 px in the images of this study, resulting in $k_{1}=0.2195$ mm/px).

From Figure 12, the apparent position of the interface points is connected to the distance between the camera and the centerline of the pipe by:(16)$$\tan\Omega =\frac{x_{apparent}}{L}$$
where tanΩ is very small, since the camera is far from the pipeline ($R_{out}/L\ll 1)$, and it can be approximated via tanΩ≈sinΩ≈Ω and cosΩ≈1. The approximation allows writing Equation (11) as:(17)$$x_{real}\approx \frac{1}{\cos\left(\phi_{1}-\phi_{2}+\phi_{3}-\phi_{4}\right)}\frac{n_{air}}{n_{water}}x_{apparent}$$

Moreover, the approximation allows combining Equations (14) and (15) into:(18)$$\sin\phi_{1}\approx \frac{x_{apparent}}{R_{out}}$$

Closing the problem of correcting each interface point based on Equations (10), (11), (14), (17), and (18), where the angles are calculated departing from $\phi_{1}$, obtained via Equation (18), $\phi_{2}$ via Equation (10), $\phi_{3}$ via Equation (14), $\phi_{4}$ via Equation (11) and, finally, $x_{real}$ via Equation (17).

Now that the upper and lower boundaries of the gas core (represented as red in Figure 11c) are corrected for every single column of pixels along with an image, the local core diameter (for one column of pixels) can be calculated assuming that the gas core at the location forms a perfect disk, with its diameter given by the distance between the 2 points. Then, the void fraction of gas in one image can be calculated integrating the local void fraction along with the image, as:(19)$$\alpha_{cam}=\frac{1}{n}\underset{i}{\sum }\frac{d_{i}^{2}}{D_{in}^{2}}$$
where $d_{i}$ is the core diameter in the i-column of pixels.

The equivalent diameter of the core for both the ERT and camera is compared during the results section. The equivalent diameter of the core is obtained by conserving the volume of the air present in the location but assuming a cylindrical core. Then, the equivalent diameter is calculated by:(20)$$d_{eq}=D_{in}\sqrt{\alpha }$$

## 3. Results

### 3.1. Static Measurements

The proposed image reconstruction algorithms were first evaluated, performing static measurements to analyze the accuracy of the camera and ERT reconstruction algorithm. The experimental protocol was introduced in Section 2.2.

#### 3.1.1. Camera Results

The phantoms are first used to validate the accuracy of the refraction correction proposed for the camera images. Figure 13 presents a comparison between the camera values obtained via the refraction correction and the real sizes of the phantoms.

The results obtained point to a maximum error of 6% when measuring the phantoms using the camera images, with a typical overestimation of the value. Possible reasons for the deviation are: (i) an imperfect positioning of the phantom in the center of the pipe, (ii) approximation of the acrylic properties based on literature values instead of measured, (iii) impurities in water, and (iv) distortions in the camera image when compared to reality (caused by the curvature of the lens, small residual angles when positioning the device). It is worth mentioning that there is an overestimation of up to 40% in the core size without the refraction correction, making the approach fundamental when comparing any ERT results with flow imaging.

Thanks to the satisfying results of the camera in predicting the size of the phantoms, the remaining plots are done against the value to keep the same reference (the camera) for all measurements (static and dynamic). However, it is emphasized that, as the gas core is not a perfect cylinder, and the camera image processing algorithm is relatively sensitive to the passage of bubbles in the flow, the values correspond to a guideline of what is expected inside the pipe when handling dynamic ERT measurements.

#### 3.1.2. ERT results

For each phantom, 10 s of data was recorded, and by taking the average of each data, image reconstruction was done. The reconstructed images from both LBP and GN image reconstruction algorithms are shown in Figure 14. The red color in each image represents the phantom, and the water phase is represented by green color.

Images reconstructed both from LBP and GN algorithms provide good quality sharp images, and both phases can be distinguished accurately. By visually comparing the reconstructed images with the simulated images of each case, LBP is overestimating the diameter and also producing artifacts. There also exists an overestimation of the phantom sizes when using GN, but less than what has been noticed with LBP. The behavior can be verified after processing the images. Figure 15 presents a comparison between the diameters calculated using both reconstruction schemes, considering the average and binary approaches, and the camera values. It is notable in Figure 15 that every approach overestimates a lot the size of the phantoms. Naturally, as the reconstruction of the images tends to work better for more prominent elements that reach locations closer to the electrodes, the error tends to decay with increasing the size of the phantom. In both the images of Figure 14 and the curves of Figure 15, it is notable that the sensor presents a low but relatively stable (linear) gain when increasing the phantom size for the range studied.

Among the curves obtained, the LBP reconstruction presents worse results when compared to the GN scheme. Due to the better behavior of the GN reconstruction scheme, combined with the low increase of computational cost by using the method when compared to LBP, its results are extended to the dynamic study.

### 3.2. Dynamic Measurements

The averages of the ERT and camera dynamic measurements for the 25 experimental conditions adopted are plotted against each other in Figure 16, for both the averaged and binary Gauss-Newton approaches. The results obtained show that although the binary approach typically results in a smaller divergence between the two measurements, there are still errors above 100% in tracking the core, and the data presents a significant scattering of values, especially for small cores. On the other hand, the averaged approach results in a nice curve trend connecting the camera and ERT measurements that seem to follow the static values, which can be used to correct the measurements.

The quadratic curve fitting the static measurements in Figure 16a is given by:(21)$$d_{eq\;cam}=-1.511\;d_{eq\;ERT}^{2}+152.8\;d_{eq\;ERT}-3794$$
where the dimensions of the core measured by the techniques are in mm, the coefficient of determination *R*^2^ is equal to 0.9914, which indicates a very good fit.

The same trend observed for the static points of Figure 16a seems to hold for the dynamic points that are located close to the blue line. Then, Equation (21) was used to predict the camera diameters that should take place in the pipe based on the dynamic ERT points measured. Comparing the predicted values with the measured values by the camera (red points in comparison to the blue dashed line) results in $R^{2}=0.8761$, which points to a relatively good approximation of the relation by Equation (21) even when the dynamic points are studied.

Due to the fixed trend of the average curve and the good fit obtained for the dynamic points, a correction of the ERT measurements is proposed based on the fitted curve obtained for the static tests, i.e., the reconstructed image of the ERT is processed into a diameter ($d_{eq\;ERT})$ and (21) is used to convert the value to an expected core diameter took place inside the pipe and observed by a camera. The curve acts as a calibration of the sensor to the camera, where the “real” ERT equivalent diameters are obtained from the (average) measured distribution via Equation (21). It is emphasized that the expression was obtained for phantoms between 10 mm and 40 mm diameter size, and any value outside the range consists of extrapolation. Then more phantoms with larger sizes are required to confirm the trendline adopted.

The corrected equivalent gas core diameter obtained for the averaged GN reconstruction method is plotted against the camera values in Figure 17. The result mainly shows an underestimation of the core size w.r.t. the camera results (after the refraction calibration), with a deviation of less than 20%. When considering the limitations of the calculations based on the camera images and its approximations, the extrapolation of the calibration curve, the distance between the fast camera and ERT measurement locations, and the initial error of up to 300% in the ERT measurements without the calibration, the corrected results are exceptionally good.

Not only the average behavior is essential, but the recalibrated sensor should be able to capture reliable instantaneous void fractions/gas core diameters for the proper monitoring of the process when controlling the separator. As briefly mentioned in Section 2.4.2, it was not possible to synchronize the fast camera and the ERT measurements, as the first was manually activated, and both measurements present a random behavior, not allowing cross-correlation of both signals to find the time shift between the two measurements. Hence, the measurement time series is performed by analyzing the Probability Density Functions (PDFs) of the camera and ERT signals.

As the gas core equivalent diameter is in the interval between zero and the inner pipe diameter (84 mm), bounded distributions must be used to evaluate the statistics of the core. In this work, the beta distribution was chosen to provide the PDF of histograms obtained from the camera and the ERT. When the values of the equivalent diameters are divided by the inner pipe diameter, the beta distribution is defined between 0 and 1 and normalized [56].

Figure 18 presents the beta distribution fitted to the histograms of experimental point 11. During the plot of the histograms, the normalized $d_{eq}$ values from the 120 last frames of the ERT measurement, matching the 10 s of the camera recording, are plotted. For the camera, all the 500 normalized $d_{eq}$ values extracted from the flow images are used to build the histograms. It was observed that the beta distribution matches well the histograms obtained for all the 25 experimental points studied. The Beta distributions of all the cases studied are presented in Figure 19, and their respective statistical data (mean and standard deviation) is shown in Table A2
Appendix A.

It is evident from Figure 19 that the ERT typically presents a broader distribution of gas core diameters than the camera, which results in a smaller maxima value. The main divergence in the results occurs for the experimental points 3–5, where the gas core is unstable, continuously losing symmetry and breaking into bursts of bubbles. This behavior is not only a challenge for the image processing of the camera images, where the bubbles are miscategorized as part of the core, and the hypothesis of circular gas core profile does not hold, but also for the ERT measurements, where the gas phase touches the electrodes of the sensor causing overestimations of the phase distribution in the pipe cross-section. Moreover, the technique is not sensitive enough to detect small bubbles present in the flow. Consequently, both measurement techniques for such points fail in representing the reality instantaneously. However, both curves tend to present a similar average value.

The behavior is maintained to some extent in the experimental points 9 and 10, although a better match of the curves was achieved. This is a consequence of the increase in the liquid amount in relation to the experimental points 3–5. It was observed that the increase in the water amount shrinks and stabilizes the gas phase in the center of the pipe, and a stable core is observed for the remaining experimental points. This is reflected in the PDFs that present a good match between the ERT and camera calculated diameters, except for point 20, where no explanation was found for the observed divergence between the measurements. During the experiments and data processing, it was observed that at higher liquid velocities varying gas have a small effect on the size of the gas core, thus reaching a point of saturation. Most stable gas cores in terms of size and shape can be obtained at the liquid velocities higher than 1 m/s for the swirl element used in this study. Last but not least, one can note that the two beta distributions are the closest when the V_sl_ is greater than 1 m/s, and V_sg_ is lower than 0.4 m/s. This is visible in Figure 19 (experimental points 11–13, 16–18 and 21–23), and data are shown in Table A2, Appendix A.

The breakage of the vortex for the experimental points 3–5 is confirmed in the 2.5D ERT image reconstructions of Figure 20. In the Figure, the gas core is represented in orange, whereas the artifacts produced by the breakage and bubble touching the electrodes are in blue. In Table A2, Appendix A, it is also evident that the standard deviation (STD) of GN is considerably larger than the camera in the region of low superficial velocities is also due to the fact when the breakage occurs, and the gas core touches the electrodes. Also, the camera images are averaged over 0.5D (40 mm), while the ERT has a more independent measurement, which allows ERT to have a more accurate estimation of the shape. This averaging damps oscillation, which reduces the standard deviation of the data.

The average projections of all the studied experimental points are shown in Figure 21. It can be noted that the superposed gas core tends to be a circular shape in most of the cases, which explains why the average values tend to match very well for both modalities, even when constant breakages of the gas core are present. The average circular behavior of the core, positioned at the centerline of the pipe, is in agreement with the hypothesis adopted during the refraction-based camera calibration.

## 4. Conclusions

In this paper, the novel idea of correcting the size obtained by ERT images for an inline fluid separator was introduced. First, the static ERT test was done with phantoms of known sizes and, the processed reconstructed images were compared with the fast camera images. A careful refraction correction for the camera images was performed using advanced image processing techniques. The working capability of the algorithm to measure the correct diameter was verified by taking static images of the same phantoms. The processed camera images presented an error below ±10%, and 300% overestimation was observed in the image reconstruction of the ERT measurements. To obtain actual diameter values from ERT measurements, a calibration based on static tests was made.

To further test the proposed approach, two-phase flow measurements were done at different gas and liquid superficial velocities, and the gas core diameter calculated by both imaging modalities were compared. The results from both camera and ERT follow the same trends of diameter change by varying the gas velocities at a fixed liquid velocity. The correction factor computed by static measurements was used to correct the gas core diameter obtained by ERT, achieving below 20% divergence in relation to the measurements departing from the fast camera. Also, the correction function obtained for static measurements also fits well the dynamic measurements, enabling its use for diameter core size estimation during IFS experiments (this reflects in the R^2^ values). PDFs of the camera and corrected ERT values for each experimental point were compared, and a good agreement between the curves was noted, although the proposed method shows limitations in measuring the instantaneous gas core size when breakages are frequent.

The PDFs study was made on cumulated data recorded for both modalities during synchronized periods of 10 s. However, no direct comparison between ERT and camera measurements was possible due to the different acquisition frame rate and measurement positions along the installation pipe. A possible improvement could consider a clock synchronization between both modality hardware.

In general, the study successfully bridges the Electrical Resistance Tomography to the measurement of the gas core size. For future studies, however, more phantoms must be used for the calibration step, and real-time measurement of the core must be implemented for control applications.

## Figures and Tables

**Figure 1 sensors-20-06069-f001:**
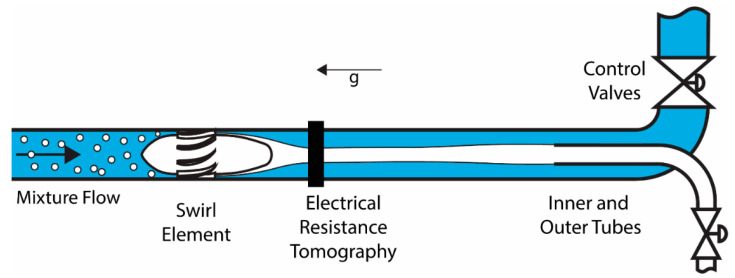
Swirl Separator Schematics. An air-water separation is represented, where the air is rendered in white and water in blue. Changes in the gas core observed at the ERT location propagate to the inner and outer tubes. The separator is installed vertical, and g represents the direction of the gravity vector in relation to the equipment.

**Figure 2 sensors-20-06069-f002:**
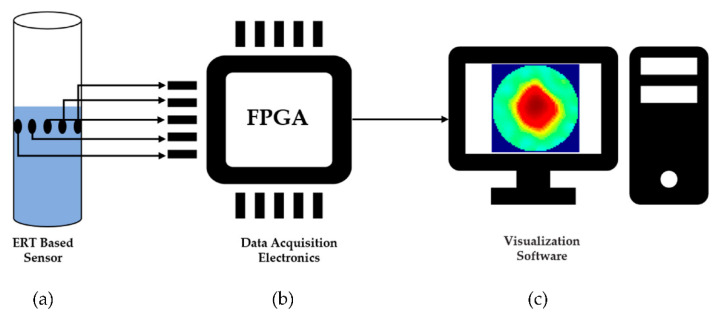
Structure of typical ERT system showing: (**a**) ERT Based sensor (**b**) Data acquisition electronics (**c**) Image reconstruction and visualization software.

**Figure 3 sensors-20-06069-f003:**
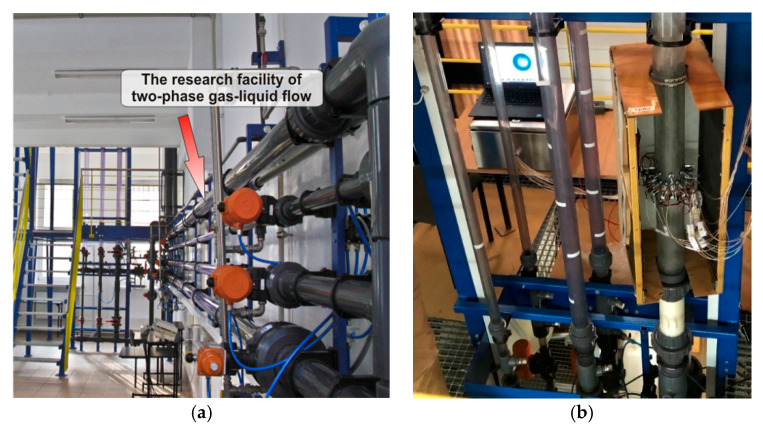
(**a**) Tom Dyakowski Process Tomography Laboratory horizontal and vertical liquid/gas Flow loop (**b**) Vertical flow installation with swirl element and ERT sensor mounted on the 90 mm pipe and surrounded by a light source cage. The Figure also shows in the back the live ERT image reconstruction module from TomoKisStudio and the Rocsole Ltd. FlowWatch device.

**Figure 4 sensors-20-06069-f004:**
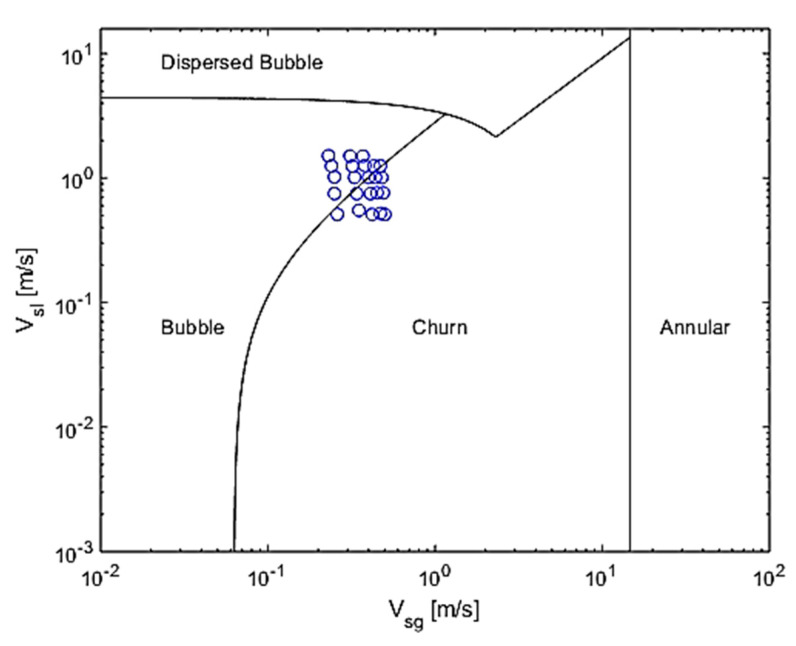
Experimental points and inlet patterns expected upstream of the swirl element, according to [36].

**Figure 5 sensors-20-06069-f005:**
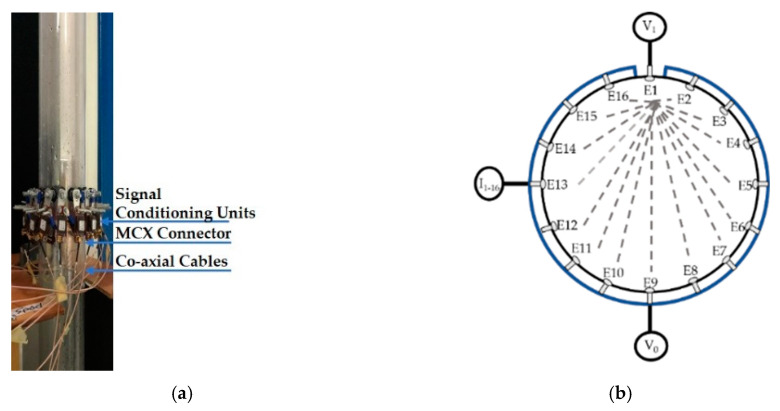
(**a**) Physical 16 electrode single layer ERT sensor placed above the swirl element (**b**) Schematics of VC-based ERT data acquisition system.

**Figure 6 sensors-20-06069-f006:**
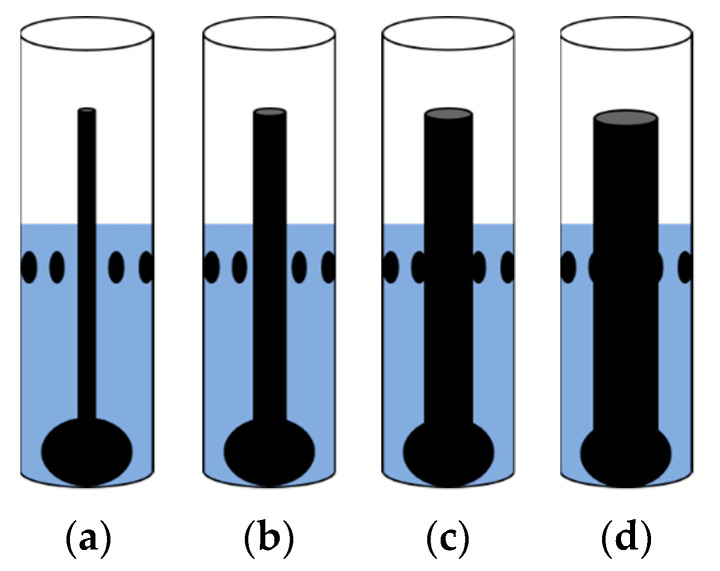
Schematic of ABS phantom placement middle of the sensor: (**a**) 10 mm (**b**) 20 mm (**c**) 30 mm (**d**) 40 mm.

**Figure 7 sensors-20-06069-f007:**
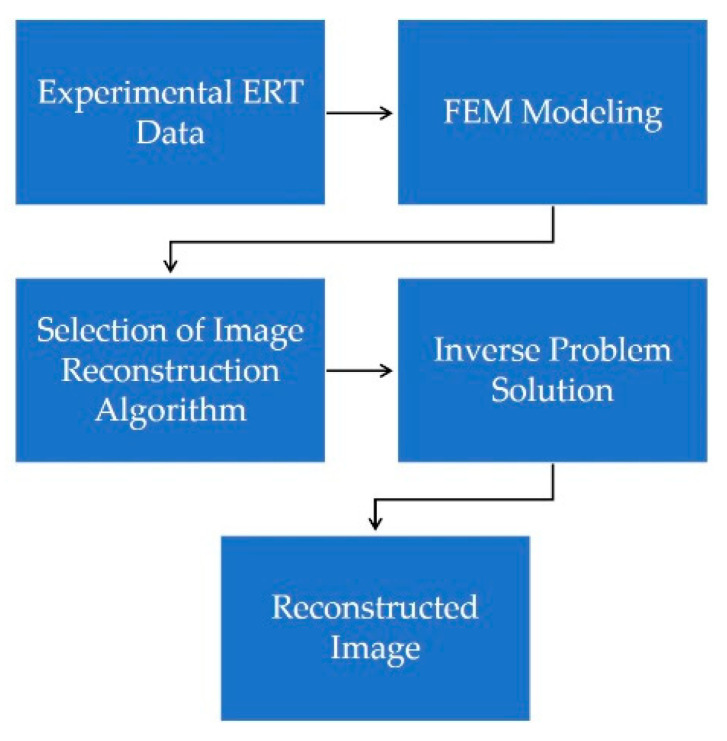
Image reconstruction procedure followed by EIDORS.

**Figure 8 sensors-20-06069-f008:**
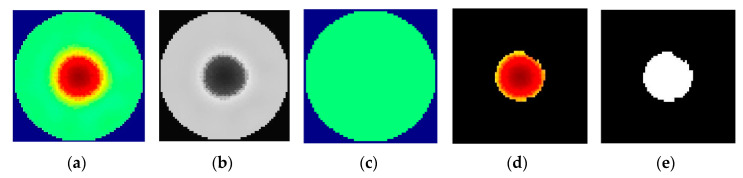
Image Processing Scheme: (**a**) Original image (**b**) Grayscale image (**c**) Background (**d**) Foreground (**e**) Segmented image.

**Figure 9 sensors-20-06069-f009:**
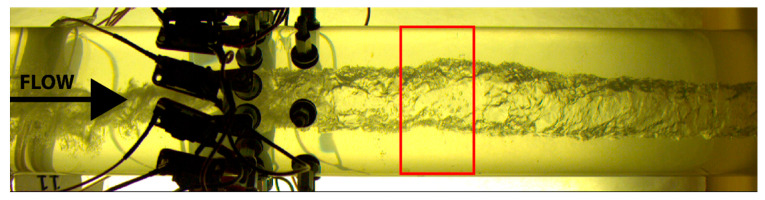
Original Image (2040 × 500 px) and box considered during the image processing and core size calculation. The obtained image after the cropping has a size of 200 × 400 px. The image corresponds to the 26th frame of the experimental set 11.

**Figure 10 sensors-20-06069-f010:**
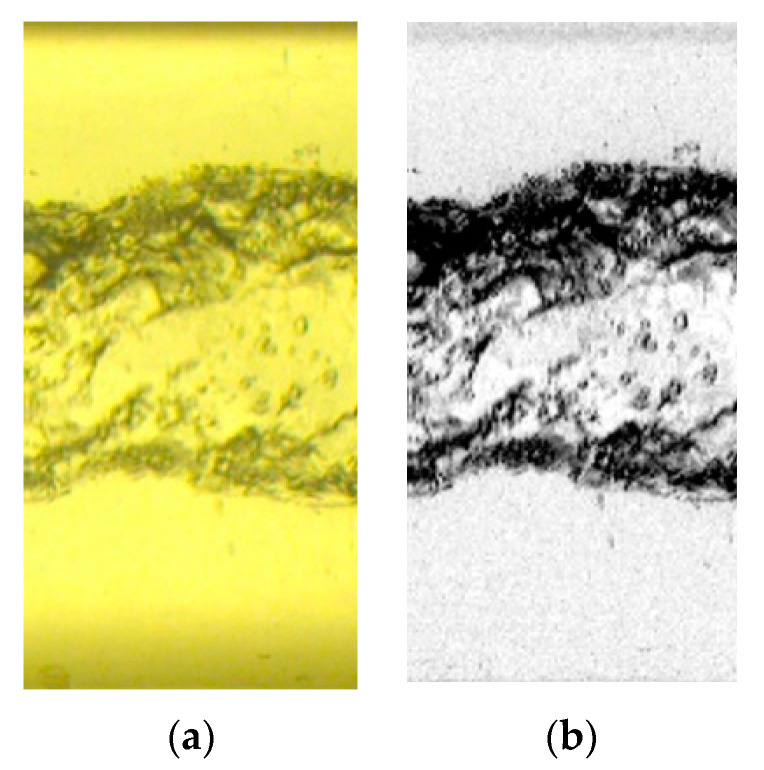
Pre-processing of the flow images. (**a**) Cropped box from the original picture. (**b**) 8-bit contrast-enhanced result.

**Figure 11 sensors-20-06069-f011:**
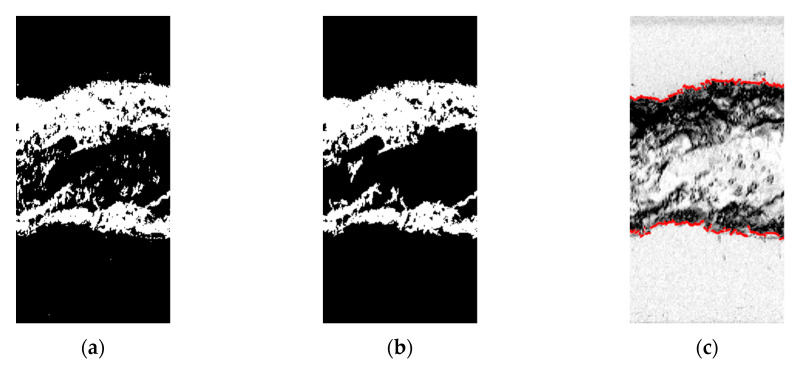
MATLAB image processing steps. (**a**) Binarized flow picture after color inversion. In the image, the interface between the fluids is represented in white and has value 1, and the locations inside the gas core and the water annulus are represented in black, with value 0. (**b**) Image obtained after the removal of small (smaller than 100 px) white regions required for proper tracking of the interface between the fluids. (**c**) interface tracked and plotted in red on top of the original image.

**Figure 12 sensors-20-06069-f012:**
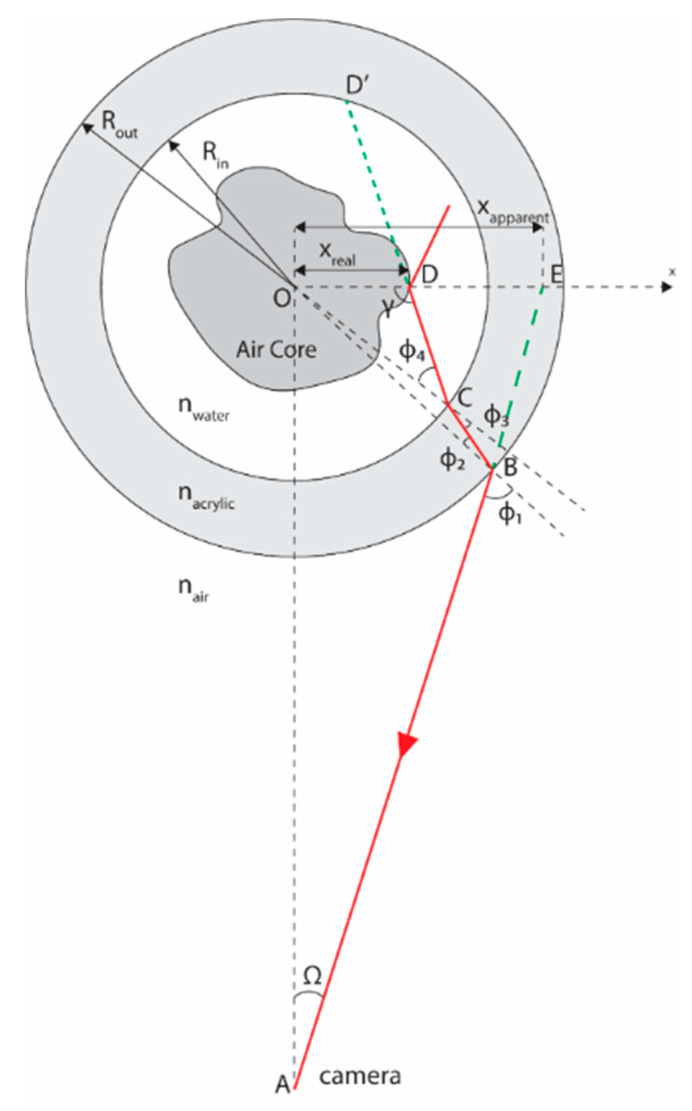
The refraction model is based on light rays reaching the camera. Their red line corresponds to the light ray, and the green lines correspond to extensions of the light rays that are useful for the analysis. The *x*-axis is placed on top of the focal plane of the camera, and the light ray path is for illustration porpoise only, without taking into account the real scales of the setup or the actual path of the reflected light ray.

**Figure 13 sensors-20-06069-f013:**
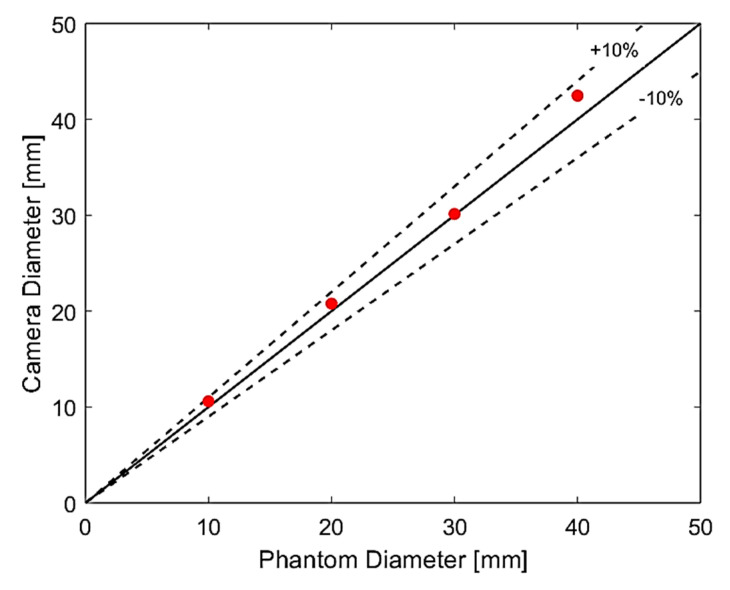
Comparison between phantom sizes and values recovered from camera images. The measured points are plotted in red, the continuous line represents a perfect match between the camera and the phantom size, and the dashed lines represent ±10% deviation lines around the average.

**Figure 14 sensors-20-06069-f014:**
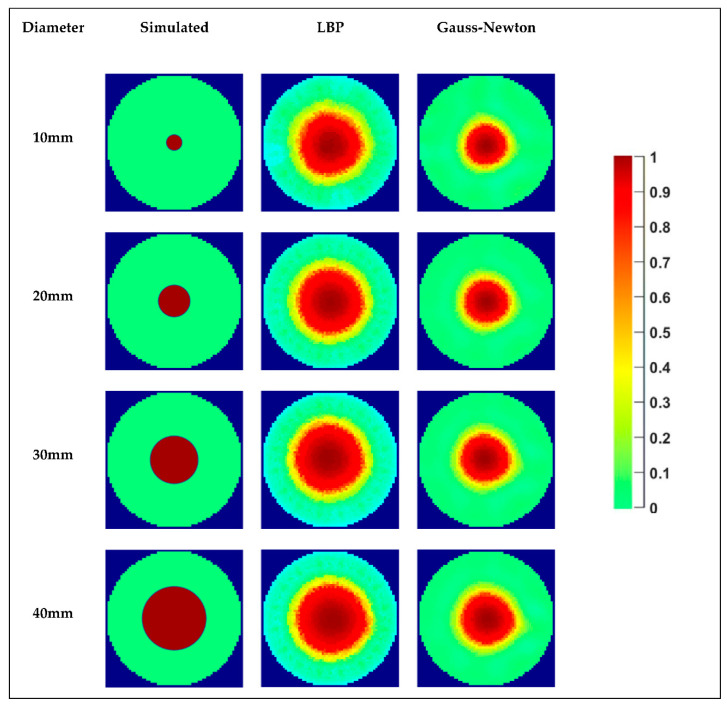
Image Reconstruction based on static tests: Row 2 (Column 2-Column 4) Case 10 mm: Simulated image, LBP reconstruction, and GN reconstruction. Row 3 (Column 2–Column 4) Case 20 mm: Simulated image, LBP reconstruction, and GN reconstruction. Row 4 (Column 2–Column 4) Case 30 mm: Simulated image, LBP reconstruction, and GN reconstruction. Row 5 (Column 2–Column 4) Case 10 mm: Simulated image, LBP reconstruction, and GN reconstruction. Column 5: Colormap.

**Figure 15 sensors-20-06069-f015:**
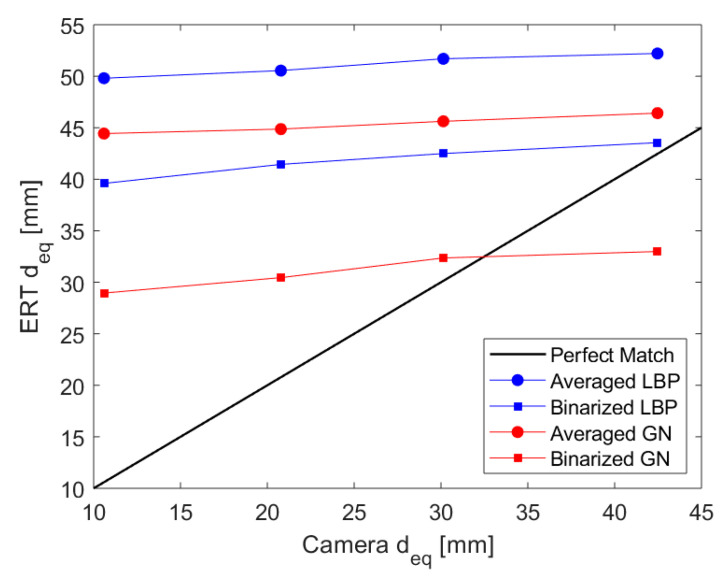
ERT reconstructed diameters plotted against the phantoms’ diameters obtained from the fast camera. The Linear Back Projection algorithm is represented in blue, with the average method presenting circle marks and the binary method presenting square marks. The results for the Gauss-Newton Reconstructed images are presented analogously but in red. The black line in the image provides a reference for a sensor mimicking the camera (parity of values).

**Figure 16 sensors-20-06069-f016:**
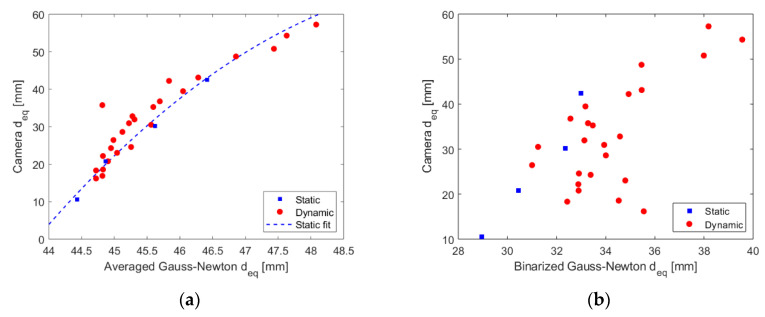
Average equivalent diameters obtained using the camera plotted against the Gauss-Newton reconstruction scheme for the static and dynamic experimental points. (**a**) Averaged Gauss-Newton scheme. The blue dashed line of the graph is obtained by fitting the 4 static points but plotted for the entire range of dynamic data for comparison. (**b**) Binarized Gauss-Newton scheme. As the dynamic data does not follow a clear trend in the Figure, no fit based on the static measurements is presented.

**Figure 17 sensors-20-06069-f017:**
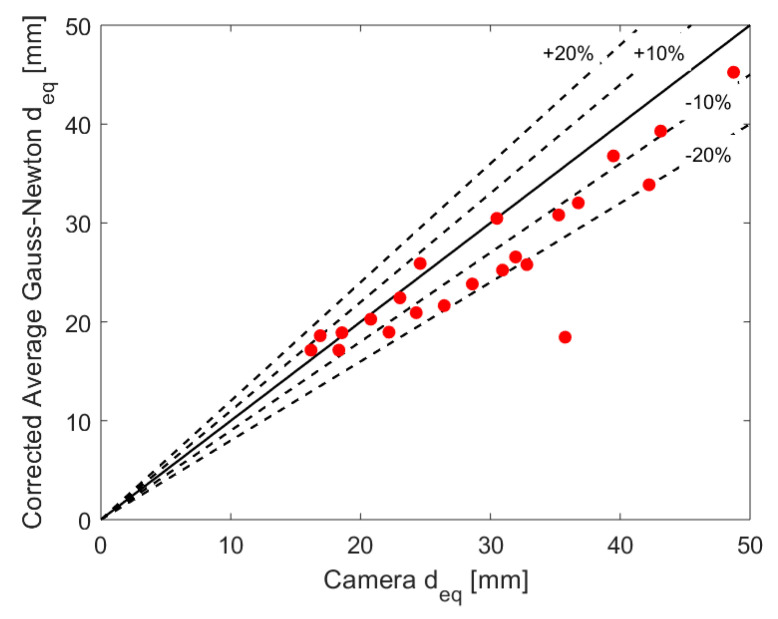
Calibrated gas core average equivalent diameter provided by the Averaged GN scheme plotted against the camera values.

**Figure 18 sensors-20-06069-f018:**
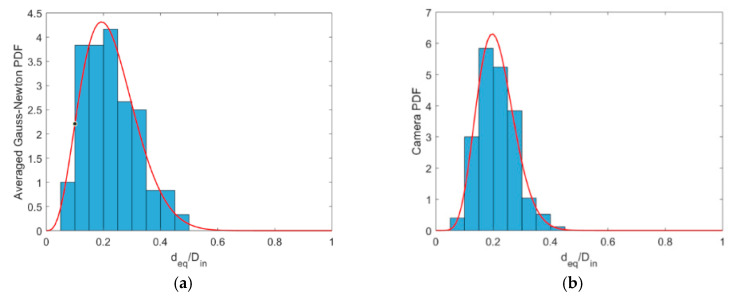
Histograms and Beta distribution PDFs for the ERT (**a**) and Camera (**b**) values for experimental point 11. Only the 120 frames of the last 10 s of ERT measurement are considered during the plot. Both histograms are plotted in 20 intervals uniformly spaced between 0 and 1.

**Figure 19 sensors-20-06069-f019:**
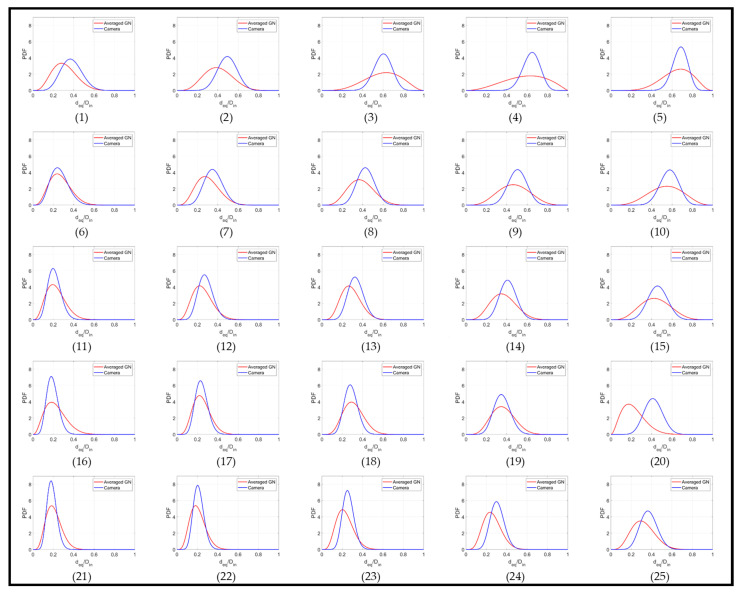
Probability Density Functions for the equivalent core diameter of each experimental point studied. Camera Beta distributions are plotted in blue, while the corrected ERT distributions are plotted in red.

**Figure 20 sensors-20-06069-f020:**
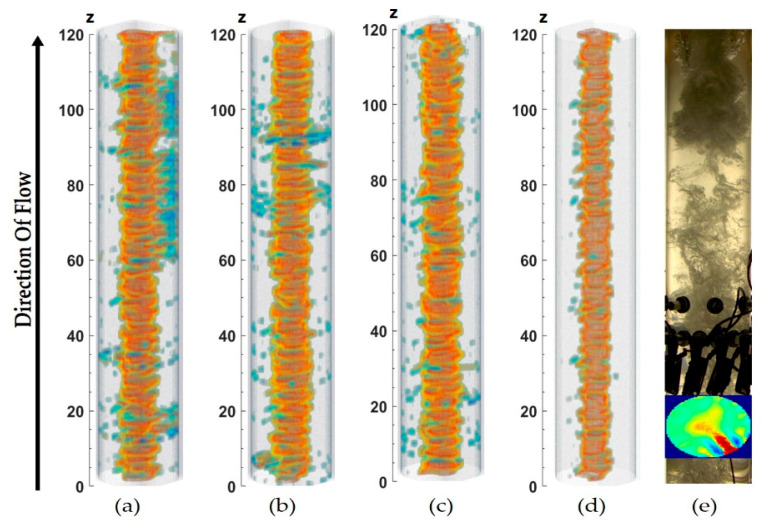
2.5D images for: (**a**) Experimental point 3 (**b**) Experimental point 4 (**c**) Experimental point 5 (**d**) Experimental point 6. (**e**) Manually calibrated ERT and camera image showing the failure of ERT due to gas core breakage.

**Figure 21 sensors-20-06069-f021:**
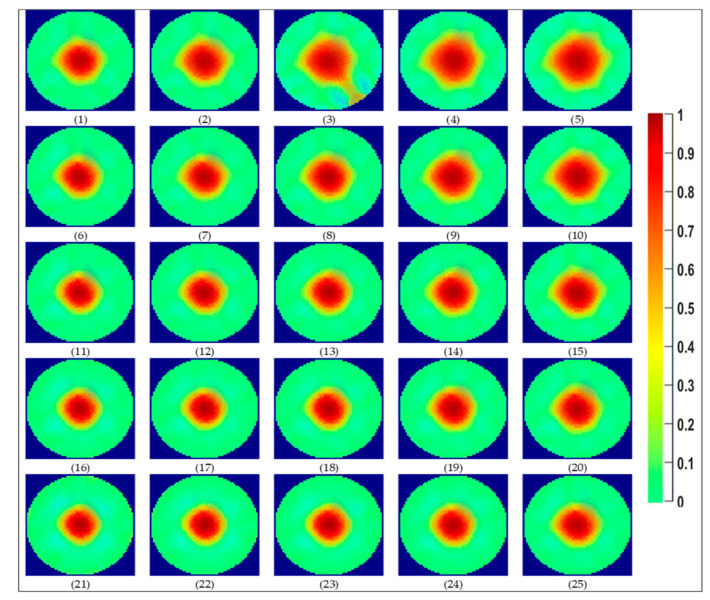
Average projection of the GN reconstructed images obtained for all experimental points. Each row shows images obtained, keeping liquid superficial velocities constant and varying the superficial gas velocities, as it can be guessed from Table A1 in Appendix A.

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
