# Peer review of "Electrical Resistance Tomography for Control Applications: Quantitative Study of the Gas-Liquid Distribution inside A Cyclone"

_sensors, 2020, doi:10.3390/s20216069_

Round 1
Reviewer 1 Report
The presented paper is very extensive and the methodology and results are described in detail. I have a following comments and recommendations:
In materials and methods
- add specific type of the camera,
- provide more information about the light source
The methodology of the static measurement may be moved to materials and method section.
Figure 16. (a) The blue dashed line should be better explained in the caption.
In line 456 is: “ When using equation (21), the dynamic points result in R2 = 0.8761”. If the eq. (21) represents fit of the static data, the R2 for dynamic points can not be calculated from this equation. I recommend use a different approach for comparison of the dynamic and static tests.
Figure 19. I think that the evaluation in this figure is not sufficient. There is space for more sophisticated approach. e.g. statistic comparison of median values, variances.
The conclusion section should contains primary conclusions and it should not be just a summarization of the paper.
I would like to note that the fact that the camera a and ERT sensor were not synchronized significantly reduces the strength of the results.
Reviewer 2 Report
This paper deals with electrical resistance tomography combined with camera capture for estimating the gas content in a liquid flowing through a pipe. The authors show that using only one of these sensors leads to estimation error while the association gives better results. The paper is well written, particularly pedagogical and argumented. I recommend this paper for publication.
I have no real corrections to the manuscript to ask to the authors, but only some comments and question:
1) You use resistance tomography which is fine for this study, but, as you mentioned, the interface resistance may evolve with time, thus reconstruction alogorithm may be impacted, particularly in the case of a non-uniform matter deposition on the electrodes. Why didn't you consider ECT which is less sensitive to such drifts ?
2) You use only one camera, so you had to assume a more or less circular shape to the gas part. With 2 or more cameras, the assumption would not have been so strong, wouldn't it?
3) In Figure 14, it is clear that the reconstruction algorithms are not very efficient to estimate the phantom radius. This may be due to the fact that you use a general algorithm while your system has less degree of freedom. With 16 electrodes you have 120 independent data, while the gas shape has, say, 1 positions (x and y), 2 axis dimentions and 1 angle (for an elliptic shape), 2 material resistances, so 7 degrees of freedom or so. You should then consider to reduce the number of degree of freedom in your calculation, so your problem would be less ill-posed, and your results would as a consequence be more accurate, I presume. Look at the work of Oussar (https://doi.org/10.1002/jnm.2245 and https://doi.org/10.1108/COMPEL-09-2015-0352), He made something of that type with good results, though it was with the ECT technique.
Reviewer 3 Report
This manuscript introduced a novel idea of correcting the size obtained by ERT images for an inline fluid separator. The concept is novel and this method shows great significance at future ERT applications.
Some questions arise related to the proposed method and some details of experiment need to be clarified.
- In the Conclusions part, the author mentioned that the error has been reduced to 20%. Is this error between ERT and the camera? Also, the souse of error is not discussed.
- On page 21, the author mentioned that “The correction factor computed by static measurements was used...” Is the correction factor represents equation (21)? Additional, how to use the correction factor to correct ERT result is not very clear, please explain this.
- In this work, high-speed cameras are used to correct the diameter obtained by ERT. However, there is still 6% error when using camera images. Would the error accumulation be considered in this work?
- Twenty-five experimental points were chosen based on steps of 5 m3/h in the flow rates of each phase, starting at 10 m3/h. Are there any restrictions on flow rates in this experiment?
- It isn’t very clear why to choose beta distribution to provide the PDF of histograms obtained from the camera and the ERT.
- In Figure 18, the unit of y-axis is not presented.
- In equation (21), the meaning of R2is not clear.
Reviewer 4 Report
The topic is relevant and challenging. The paper is well written and organized, and gives relevant results.
The paper is worth publishing in Sensor after the authors address the following comments:
1-the originality of the work should be clearly stated at the end of the introduction.
2-page 4 line 11 authors said the excitation pattern provide "120 current independent data." Actually, this statement is not accurately true, because the excitation strategy described is known to show redundancies with linearly dependent measurements. Please correct.
3-Nothing is proposed to improve the ERT reconstruction, however one could take advantages of alternative excitation strategies like the harmonic one, also iterative methods could be considered as a few number of iteration can give good results allowing the ERT to give instantaneous images. Finally, eigenvalues methods could also be considered as they can give a good estimation of void fraction.
Reviewer 5 Report
My technical background (40 years) is in systems and control, so I found this work to be very interesting.
I found the paper to be well-organized, and easy to read. Some minor English grammar corrections could be made in isolated instances, but these errors do not affect my understanding of the work. For example, on line 111, the words "current independent data" could be changed to "independent current data." Again, these are minor issues that I did not consider irritating in any way.
I do have one suggestion regarding Figure 1. There is a left-direction arrow with label "g". What is the meaning of this arrow, and the symbol "g"?
As a control systems researcher, I was expecting more work in the area of control systems. The title of the paper also has the words "Control Applications." But I found little to no discussion regarding control application issues. For example, what would be the transfer function of the sensing system? What are the frequency response characteristics? The work focuses almost exclusively on sensing. Therefore, I suggest removal of "control applications" from the title.
Overall, I found the paper to be very informative and enjoyable to read.
Round 2
Reviewer 1 Report
Authors took my comments into account and corrected the paper. The paper is ready for publication.